

# Discriminating conodont recording bias: a case study from the Nanzhang-Yuan'an Lagerstätte

Kui Wu[1,2,3], Boyong Yang[1], Bi Zhao[1], Liangzhe Yang[1], Yarui Zou[1], Gang Chen[1] and Jiangli Li[1]

[1] Hubei Institute of Geosciences, Hubei Geological Bureau, Wuhan, Hubei, China
[2] State Key Laboratory of Biogeology and Environmental Geology, School of Earth Science, China University of Geosciences (Wuhan), Wuhan, Hubei, China
[3] Hubei Key Laboratory of Resource and Ecological Environment Geology, Wuhan, Hubei, China

## ABSTRACT

The Early Triassic Nanzhang-Yuan'an Lagerstätte of Hubei Province, South China, preserves abundant marine reptiles in the uppermost part of the Jialingjiang Formation and provides detailed insights into marine organisms, including newly discovered and well preserved conodont clusters of the Family Ellisonidae. These conodont elements allow us to assess the bias introduced during the acquisition process. We examined conodont elements preserved on the bedding planes and those acquired after the acid-dissolving method to analyze their attributes and length distributions. We identified a biased preservation of different conodont elements related to their morphologies. After the acid-dissolving procedures, the bias increased, and all different elements were affected, with larger individuals being particularly prone to destruction. Among them, the P elements of Ellisonidae were the least affected, while the S elements were the most affected. This study further indicates that paleobiological interpretations based on fossil size or morphology could be obscured if the influence of post-mortem effect is ignored.

## INTRODUCTION

As nektonic marine organisms, conodont animals originated in the Cambrian and disappeared near the Triassic-Jurassic boundary (*Clark, 1983*; *Sansom et al., 1992*; *Goudemand et al., 2011*; *Martínez-Pérez et al., 2014*; *Martínez-Pérez et al., 2015*; *Du et al., 2020*). The conodont animal consists of a head, a trunk, and a caudal fin, with a feeding apparatus and two eyes attached to the head (*Briggs, Clarkson & Aldridge, 1983*; *Aldridge et al., 1993*). Its total length can reach up to several centimeters or tens of centimeters, while the length of a single conodont element is in the millimeter to micrometer range (*e.g.*, *Gabbott, Aldridge & Theron, 1995*; *Takahashi, Yamakita & Suzuki, 2019*). Due to the absence of a mineralized skeleton, conodont elements are usually the only preserved parts of conodont animals (*Takahashi, Yamakita & Suzuki, 2019*). Different conodont elements of an apparatus might exhibit completely different rates of evolution, and rapidly evolving

Corresponding authors
Kui Wu, kuiwu@cug.edu.cn
Boyong Yang, boyongyang@163.com

elements were more commonly considered and utilized for biostratigraphic correlations (*Orchard, 2007*; *Chen et al., 2016a*).

Conodont elements can be obtained in high abundance from strata though dissolution methods, making them highly applicable and important for biostratigraphic correlations and defining the geologic timescale, especially for the $P_1$ elements during the Permian-Triassic period (*e.g.*, *Shen, 2023*). To obtain sufficient conodont elements, the dissolution method has been utilized in numerous studies (*e.g.*, *Jiang et al., 2007*; *Sun et al., 2012*), including a recent report on extracting conodont elements from chert with NaOH solution (*Rigo et al., 2023*). For example, in studies of the Permian-Triassic boundary, these methods have provided plentiful paleontological, paleoenvironmental, and biostratigraphic information, greatly improving our understanding of the geological processes during this interval (*Sun et al., 2012*; *Chen et al., 2013*; *Dal Corso et al., 2022*; *Shen, 2023*). Conversely, due to limitations related to their size, morphology, preservation condition and preparation methods, fewer apparatuses or clusters have been found directly on the rock surface. However, more details about the conodont animal have been revealed through to these materials (*e.g.*, *Gabbott, Aldridge & Theron, 1995*; *Goudemand et al., 2011*; *Sun et al., 2020*).

We have known since the last century that the fossil record of conodonts can be fundamentally biased due to taphonomic processes and laboratory procedures (*Purnell & Donoghue, 2005*; *von Bitter & Purnell, 2005*). First of all, the preservation of conodont elements in the strata is influenced by their morphology, which may lead to biased fossilization of different anatomical units (*Purnell & Donoghue, 2005*; *Orchard, 2007*). Additionally, the differential destruction of elements during laboratory processes, particularly the acid-dissolving method, affects conodont data, including the numbers, dimensions (reducing size by breakage), and ratios of different conodont elements (*Von Bitter, 1972*; *Jeppsson & Anehus, 1995*; *Von Bitter & Purnell, 2005*). For example, the apparatus of Ellisonidae consists of 4 P elements, 2 M elements and nine S elements (*Sun et al., 2020*), while results after laboratory processes exhibited variable ratios of different elements (*Koike, 2016*; also see summary in the Supplementary File of this study). Furthermore, previous studies have shown that the size of the conodont element is not only related to ecological change but also to taxonomic identification (*Chen et al., 2013*; *Chen et al., 2016a*; *Ginot & Goudemand, 2019*). Hence, this basic biological trait of conodont elements has been largely investigated, although the impact of laboratory processes on conodonts size is usually not mentioned (*e.g.*, *Chen et al., 2013*; *Wu et al., 2019*; *Zhang et al., 2020*; *Leu, Bucher & Goudemand, 2019*). Specifically, as one of the three main Early Triassic conodont groups, the ellisonids have been less recognized and understood compared to the anchignathodontids and gondolellids, and they were thought to have suffered an extinction at the Smithian-Spathian boundary (*Orchard, 2007*). A recent study showed that large amounts of ellisonids were preserved in the uppermost Lower Triassic of Hubei Province, South China, suggesting that the Early Triassic records of ellisonids have been obscured by their special morphology as well as laboratory processes (*Wu et al., 2023*).

As one of the most famous areas for Early Triassic marine organisms, abundant and well-preserved fossil specimens have been found in the limestone of the uppermost Jialingjiang Formation in the Nanzhang-Yuan'an area (*Wu et al., 2023*), making it a fossil-Lagerstätte

for the latest Early Triassic geologic record in South China (*Benton et al., 2013*; *Kimming & Schiffbauer, 2024*). Hence, this Lagerstätte provides an invaluable opportunity to fully investigate the organisms and address biases encountered when interpreting the fossil details of conodonts from the geological records. Recently, abundant conodont elements of Ellisoniidae have been discovered in this section (*Wu et al., 2023*). Our study further contributes to this research by identifying conodont elements of Ellisoniidae from the bedding planes in this section. Associations of conodonts on the bedding planes serve as the most reliable archive for biological traits, unaffected by laboratory treatment (*Goudemand et al., 2012*; *Sun et al., 2020*). Through quantitative analysis of composition, size, and ratio of different elements, this study offers the opportunity to examine biases originating from both the bedding planes and the residues after the acid-dissolving method, with implications for other types of conodonts during the Early Triassic.

## LOCATION AND GEOLOGICAL SETTING

The studied Zhangjiawan section is about 25 km north of Yuan'an County, in the western part of Hubei Province, south-central China (*Wu et al., 2023*). During the Early Triassic, the South China block was located near the equator in the eastern part of the Tethys Ocean, while extensive shallow-marine deposits recorded in the North Marginal Basin of the Yangtze Platform (see Fig. 1 of *Wu et al., 2023*). To date, numerous fish fossils have been reported from the Lower Triassic of the North Marginal Basin, and two distinctive marine reptile faunas (the Nanzhang-Yuan'an fauna and the Chaohu fauna) have also been found from this region (*Benton et al., 2013*).

As a representative section of the Nanzhang-Yuan'an fauna, the Zhangjiawan section is well-exposed along a road and a quarry, with a thickness of approximately 120 m (*Wu et al., 2023*). The section outcrops vermicular limestone, limestone, dolomite, brecciated dolomite, laminated limestone, volcanic tuffs, and sandy mudstone, indicating that it belongs to the restricted platform facies (*Wu et al., 2023*). Reported marine reptiles were all found in the laminated limestone, which is about 36 m thick. A 0.5-meter-thick unit of wedge-like or lenticular-like strata, consisting of centimeter-sized thin beds, appear in the middle part of the laminated limestone (Fig. 1A), suggesting the deepest depositional environment with minimal hydrodynamic effect in this section. Recent studies have shown that the Nanzhang-Yuan'an fauna was extensively and well documented in this region, making it one of the youngest Early Triassic Lagerstätte for marginal sea animals, particularly those with hard skeletons (*Yan et al., 2021*; *Wu et al., 2023*; *Kimming & Schiffbauer, 2024*).

## MATERIALS AND METHOD

The materials from the bedding planes were found through systematical collection rather than incidental discovery. Bulk samples, each weighing approximately 5 kg, were initially collected from the Zhangjiawan section (*Wu et al., 2023*). These samples were then crushed into pieces measuring around 3×3 cm (sometimes lager) and processed with 10% diluted acetic acid. A conodont cluster was obtained from the residues after the acetic acid dissolving

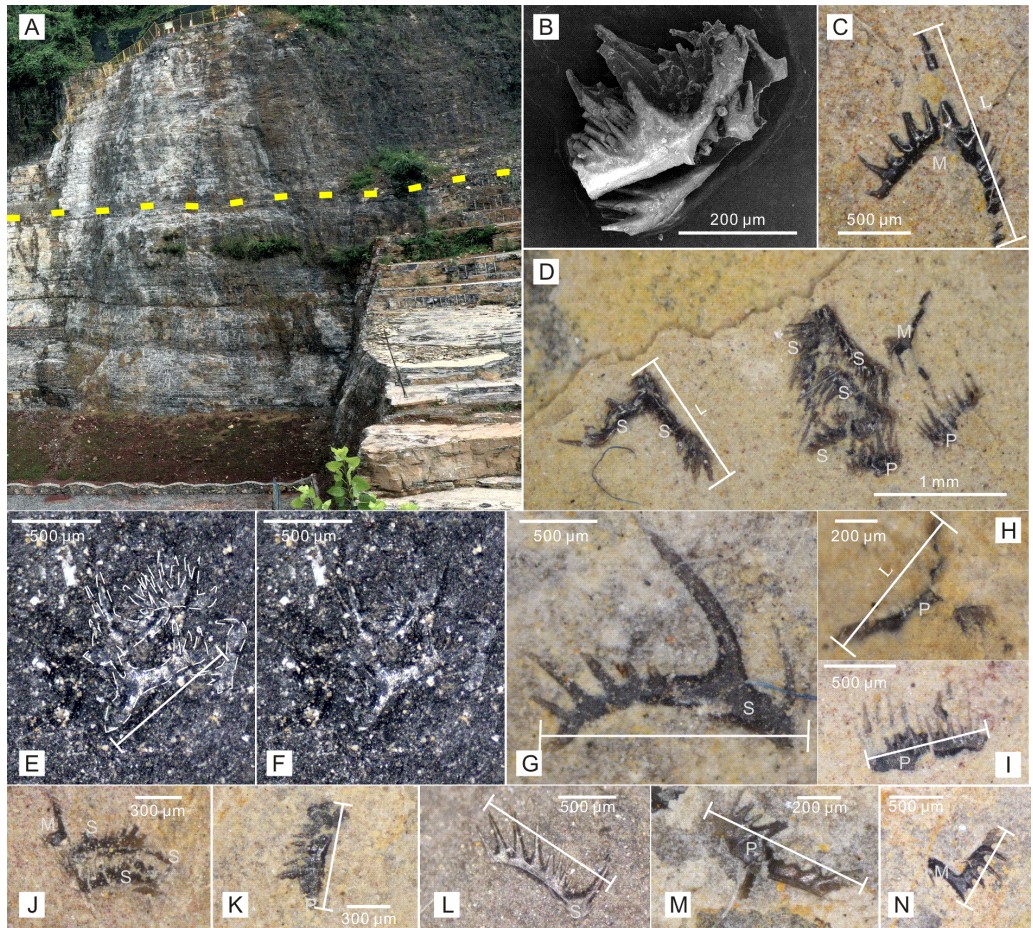

**Figure 1 Conodont elements recovered from Zhangjiawan section, Yuan'an County, Hubei Province, South China.** (A) Dark-colored laminated limestone of the uppermost Jialingjiang Formation. The dashed line indicates the thinnest beds where the clusters were found. (B) Recovered conodont cluster after acetic acid dissolution. (C) Isolated conodont element found from the bedding plane. (D, E and F) Conodont natural assemblage found from the bedding plane. (G–N) Different isolated elements and clusters found from the bedding plane. (Also see the supplementary material of *Wu et al., 2023*). Photo credit: Kui Wu.

and sieve-separating procedures (Fig. 1B), indicating that well-preserved clusters may have been preserved on the bedding planes, which were millimeters in thickness.

The sample containing conodont clusters was taken from the middle part of the dark-colored lamellar limestone, which is the thinnest bed in the Zhangjiawan section. Consequently, approximately 30 kg of cracked rocks were collected from this bed and observed directly under a binocular microscope. For better comparison, a sample weighing about 20 kg was collected from the location where the cluster was found. This sample consisted of limestone laminae, millimeters in thickness. To avoid crushing, which might destroy conodont elements, these limestone laminae were processed directly with 10% diluted acetic acid. The sample was kept in the diluted acetic acid for about 24 h until only minor or no bubble were visible. The supernatant liquid was then poured out, and fresh diluted acetic acid was added. Every 5 days thereafter, the undissolved residues were sieved

using 20-mesh (0.850 mm, on top) and 160-mesh (0.095 mm, on bottom) sieves. This process continued until all the rocks were dissolved. After drying the residues in an oven at 30 °C, they were examined under a binocular stereo-microscope to obtain conodont elements.

The lengths of conodont elements from the bedding planes and those obtained through the acetic acid dissolving method (including both complete and broken elements) were measured in microns. Following the common practice in size studies (*Wu et al., 2019*; *De Baets et al., 2022*), all data were also logarithmized (base 10) for statistical analysis. Due to their prominent cusp and the presence of the third process, Ellisonidae elements exhibit more variable morphology than other Early Triassic conodonts (*Orchard, 2007*). According to anatomical standards and morphological aspects, conodont elements were classified into three types: P, M, and S elements (*Purnell, Donoghue & Aldridge, 2000*; *Sun et al., 2020*). For M elements, the distance from the tip of the cusp to the distal end of the longer process was measured (Fig. 1C). For P and S elements with only one process, the distance between the two distal ends was measured (Figs. 1D–1G, 1I–1L). For those elements with three processes, the two longer processes were chosen, and the elements distal ends was measured (Figs. 1H, 1M). Broken conodont elements were also measured in terms of their maximum linear dimension (Figs. 1C–1D, 1M, 1N). Due to the restricted information available for Ellisonidae elements preserved on rock surfaces, further classification into P ($P_{1-2}$) and S ($S_{0-4}$) elements was not considered in this study. To make a better comparison, multielement composition data of various Ellisonidae species from *Koike (2016)* were also collected.

## RESULTS

Due to the low abundance of conodonts from the upper Lower Triassic, particularly from the Jialingjiang Formation of South China (*Zhao et al., 2013*; *Wu et al., 2023*), a total of 167 and 71 conodont elements (including both broken and complete elements) were acquired from the bedding planes and the residues after acid-dissolving, respectively (Table 1). The conodonts from the bedding planes comprised 25 P elements (14.97%), 21 M elements (12.57%) and 121 S elements (72.46%) (Fig. 2A). In contrast, the residues after acid-dissolving yielded 17 P elements (23.94%), 17 M elements (23.94%) and 37 S elements (52.11%) (Fig. 2A), indicating that the latter method resulted in fewer acquisitions of all element types. Compared with the standard composition of the Ellisonidae apparatus, M elements obtained from the acid-dissolution method and S elements preserved on the bedding planes exhibit an increase of the ratio (Fig. 2B). A comparison with the data from *Koike (2016)* suggests that results could be influenced differently due to their varying morphologies, even within the same species from different samples (Fig. 3).

Percentages of complete and broken conodont elements from the bedding planes and the acid-dissolving method were also different (Fig. 4). For the conodonts preserved on the bedding planes, complete elements comprised 21 P elements (84.00%), six M elements (28.57%), and 89 S elements (73.55%). In contrast, for the conodonts obtained from the acid-dissolution method, complete elements comprised of eight P elements (47.06%), 11 M elements (67.71%), and 12 S elements (32.43%).

**Table 1  Number, ratio, length range, length average of conodont elements and their differences.**

| Acquiring way | Position/type | N | P | Complete (N/P) | Broken (N/P) | Material/Original Ratio (P:M:S) | R (μm) | A (μm) | S (μm) | A* (μm) | S* (μm) | p | p* |
|---|---|---|---|---|---|---|---|---|---|---|---|---|---|
| Bedding Planes | P | 25 | 14.97% | 21/84.00% | 4/16.00% | 1.2:1:5.8/ 2:1:4.5 | 450~1,550 | 868.4 | 332 | 948 | 430.6 | 0.21 | <0.01 |
| | M | 21 | 12.57% | 6/28.57% | 15/71.43% | | 240~1,600 | 747.1 | 338 | | | 0.51 | |
| | S | 121 | 72.46% | 89/73.55% | 32/26.45% | | 290~2,810 | 999.9 | 450 | | | <0.01 | |
| Disolution | P | 17 | 23.94% | 8/47.06% | 9/52.94% | 1:1:2.2/ 2:1:4.5 | 447~1,226 | 753.8 | 190 | 701 | 228.5 | 0.21 | <0.01 |
| | M | 17 | 23.94% | 11/64.71% | 6/35.29% | | 341~1,345 | 680.8 | 255 | | | 0.51 | |
| | S | 37 | 52.12% | 12/32.43% | 25/67.57% | | 356~1,373 | 685.5 | 228 | | | <0.01 | |

**Notes.**

N, Number of speciemens; P, Percentage; R, Range of length; A, Average of length; A*, Average of length (all elements); p, p-Value for the t-tests (contrast with the same type); p*, p-Value for the t-tests (contrast with all elements); S, stantard deviation.

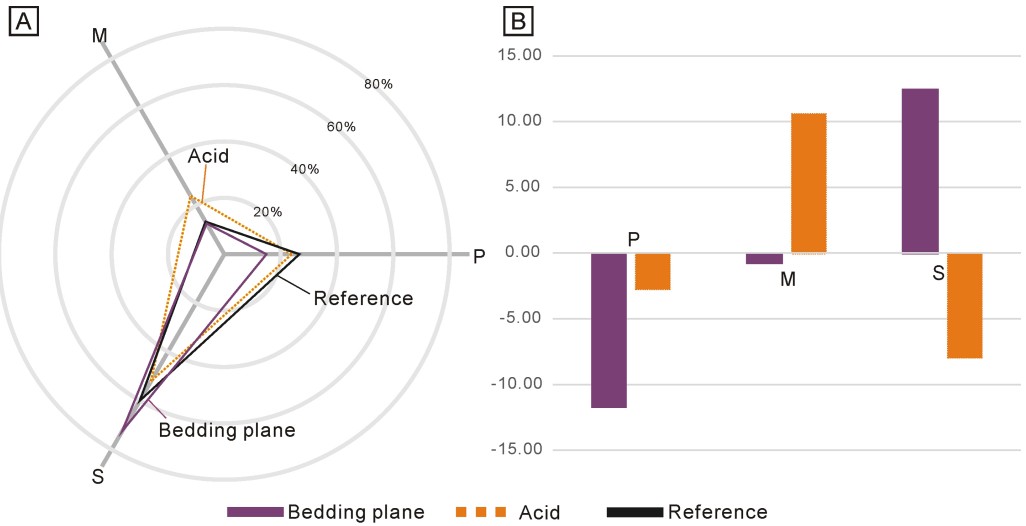

**Figure 2  Differences between the standard composition of the Ellisonidae apparatus and the materials form this study.** (A) Radar chart depicting the percentage of different conodont elements from the bedding planes, the acid-dissolution method, and the standard composition of the Ellisonidae apparatus. (B) Difference chart illustrating variations in conodont elements from the bedding planes, the acid-dissolution method, and the standard composition of the Ellisonidae apparatus. The *y*-axis represents the percentage change of different elements relative to the Reference (the standard component of the Ellisonia apparatus).

Despite the lower yield, the two groups of conodonts exhibit noticeable differences (Table 1; Figs. 5 and 6). The average lengths of the conodont elements from the bedding planes and residues are 948.4 μm and 700.7 μm, respectively, with standard deviation of 430.6 and 228.5. This suggests that conodont elements from bedding planes seem generally larger. For the conodont elements preserved on the bedding planes (Table 1), P elements range in length from 450 μm to 1550 μm, with an average of 868.4 μm and a standard deviation of 331.6 μm. M elements range from 240 μm to 1600 μm, with an average of 747.4 μm and a standard deviation of 337.9 μm. S elements range from 290 μm to 2,810 μm, with an average of 999.9 μm and a standard deviation of 449.8 μm. For the conodont

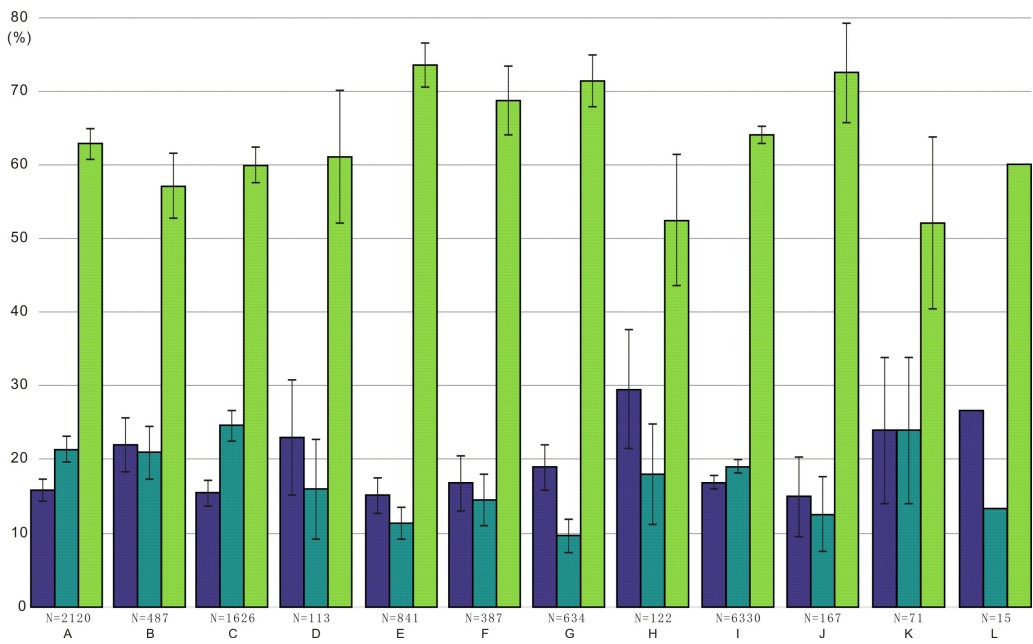

**Figure 3** **Ratios of different conodont elements from this study and *Koike (2016)* compared to the standard composition of the Ellisonidae apparatus.** Refer. represents the standard component of the Ellisonia apparatus. Error bars represent 95% binomial confidence intervals (*Raup, 1991*; *De Baets et al., 2012*). A–I are data from *Koike (2016)*. A represents Hadrodontina aequabilis (sample A), B represents Hadrodontina aequabilis (sample B), C represents Hadrodontina aequabilis (sample C), D represents Ellisonia triassica, E represents Corudina breviramulis, F represents Staeschegnathus perrii (sample A), G represents Staeschegnathus perrii (sample B), H represents Furnishius triserratus, I represents all the conodonts of *Koike (2016)*, J represents bedding plane conodont elements of this study, K represents conodont elements from acid-dissolution of this study, L represents the standard component of the Ellisonia apparatus (confidence intervals are not used here because only one apparatus is available).

elements obtained from the residues after acid-dissolving (Table 1), P elements range in length from 447 µm to 1226 µm, with an average of 753.8 µm and a standard deviation of 190.4 µm. M elements range from 341 µm to 1345 µm, with an average of 680.8 µm and a standard deviation of 254.9 µm. S element range from 356 µm to 1373 µm, with an average of 685.5 µm and a standard deviation of 227.8 µm. A two-sample $t$-test indicates that the sizes of P and M elements from different methods are not highly significantly different in size ($p = 0.51$ and $0.21$, respectively), although those from the bedding planes are generally larger. In contrast, S elements from the two groups show a highly significant size difference ($p < 0.01$). Considering all elements of different types from each group, they exhibit significantly different length distributions. A percentile plot reveals that S elements from the bedding planes include noticeable larger individuals, whereas the other types have similar length distribution percentages. Over all, conodont elements from the bedding planes tend to be larger and have a higher percentage of S elements (Figs. 5 and 6).

The length data are better distributed after logarithmisation (Figs. 7 and 8). Conodont elements from the bedding planes are generally larger than those from the residues after the acid-dissolving method (Fig. 7), the same conclusion can be drawn when the elements

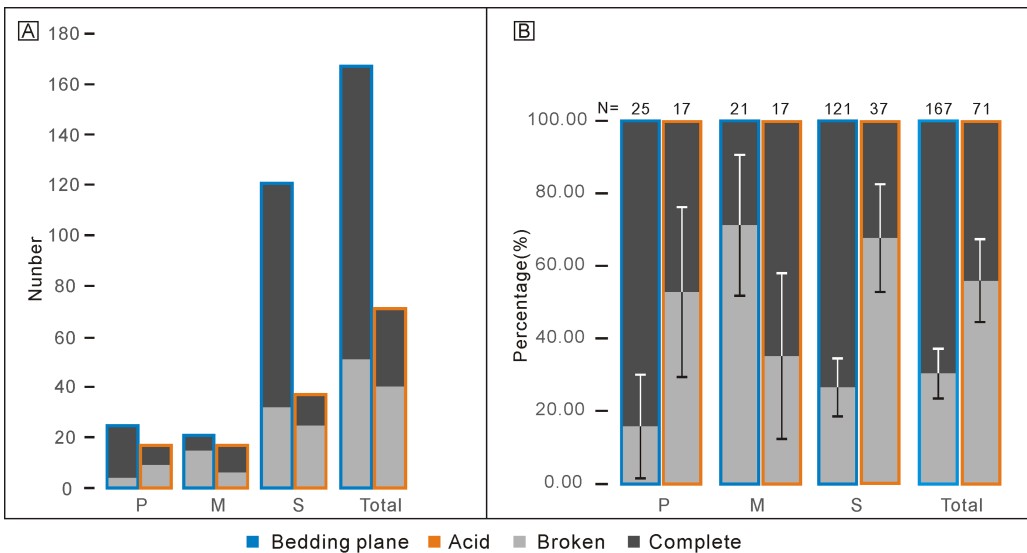

**Figure 4  Comparison of complete and broken conodont elements from the bedding planes and the acid-dissolving method.** (A) Numbers of different elements. (B) Ratios of different elements. Error bars represent 95% binomial confidence intervals (*Raup, 1991*; *De Baets et al., 2012*).

are further divided into P, M, and S elements (Fig. 8). After removing the data of broken conodont elements, the violin plots of the length suggest that their distribution modes from the acid-dissolving method have been affected more than those from the bedding planes. This is reflected by positive skewness, flat kurtosis, and smaller mean and median sizes (Fig. 9).

## DISCUSSION

Conodont elements are phosphatic micro-fossils (millimeter to micrometer) that belong to extinct marine crown vertebrates (*Donoghue & Purnell, 1999*; *Goudemand et al., 2012*). They were self-repairable if damaged when the conodont animals were alive, but they can be easily damaged after the death of the conodont animals and during extraction from the rock (*Von Bitter & Purnell, 2005*). Furthermore, post-mortem conditions, such as sediment compaction and diagenesis, may differently bias the preservation of various elements in the apparatus (*Von Bitter & Purnell, 2005*; *Purnell & Donoghue, 2005*).

The studied conodont elements were acquired from the Zhangjiawan section, which has been reported as a representative section for the Lower Triassic Nanzhang-Yuan'an Fauna (*Yan et al., 2021*). In this section, dark-colored lamellar limestones with abundant microbial-induced sediment structures and marine reptile fossils are intercalated with massive dolomites and sandstones (*Wu et al., 2023*). The acquired conodont materials are from the middle part of the dark-colored lamellar limestone, which is also the thinnest bed of the Zhangjiawan section, suggesting that these conodont materials were deposited in a low-energy environment where sorting and selective destruction had only a slight influence on their preservations. However, the co-existence of conodont natural assemblages and

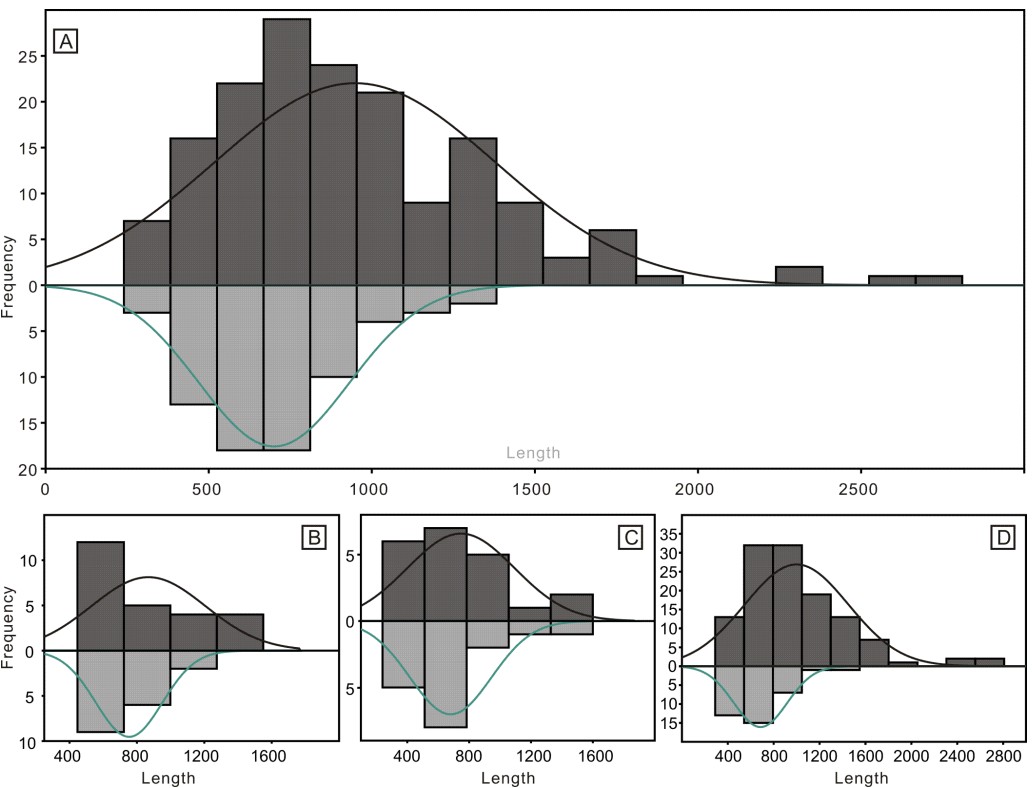

**Figure 5** **Histograms of the length of all conodont elements from the bedding planes and the acid-dissolution method.** The dark-black represent conodont elements from the bedding planes; the grey-black represent conodont elements from the acid-dissolution method. (A) All elements. (B) P elements. (C) M elements. (D) S elements.

isolated conodont elements on the bedding planes also may reflect that conodont elements experienced limited but non-negligible disturbances after their death.

The ratios of different types of conodont elements from the bedding planes and the residues after acid-dissolving indicate that those elements have been affected by both natural and artificial processes (Table 1). On the one hand, elements show different resistances to post-mortem sorting, sediment compaction and diagenesis. As a special Early Triassic group with morphological similarity between their $P_1$ and $P_2$ elements, the conodont apparatus of Ellisonidae consists of 15 elements: four P elements, two M elements and nine S elements (*Koike, 2016*; *Sun et al., 2020*). However, the conodont elements analyzed in this study from the acid-dissolving method exhibit an enrichment of P elements or a shortage of M and S elements. This suggests that conodont elements are biasedly preserved even under low-energetic water, or that they may have been differently affected by lithification (*Cooper et al., 2006*; *Sessa, Patzkowsky & Bralower, 2009*; *De Baets et al., 2022*). For example, clusters of the earliest Triassic conodont *Hindeodus* indicated that their $P_2$ elements were more difficult to access or preserve even in a deep-water environment (see *Zhang et al., 2017*) and their comments by *Agematsu, Golding & Orchard, 2018*). In shallow-water environments, stronger hydrodynamics usually resulted in the depletion of all conodont elements except

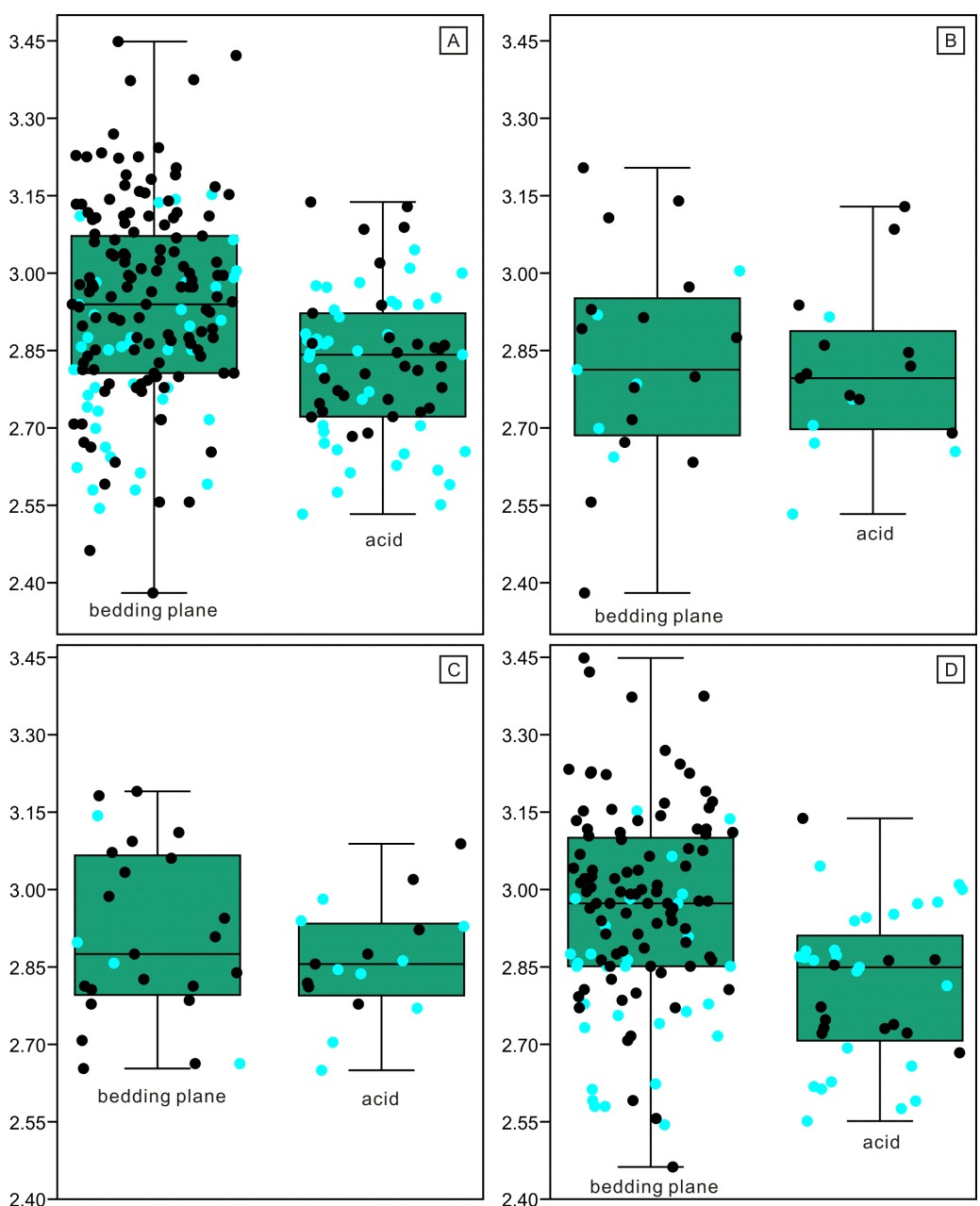

**Figure 6 Length distributions (logarithmized with base 10) of complete and broken conodont elements from the bedding planes and the acid-dissolution method.** The black dots represent complete conodont elements; the green dots represent broken conodont elements. (A) All elements; (B) M elements; (C) P elements; (D) S elements.

for the robust elements of Ellisonidae (*Jiang et al., 2014*; *Wang et al., 2023*). On the other hand, in our material, elements exhibited varying degrees of resistance to sorting during the laboratory process of the acid-dissolving method, often being broken. Compared to the conodont elements acquired from acid-dissolution, the ratio of S elements shows a

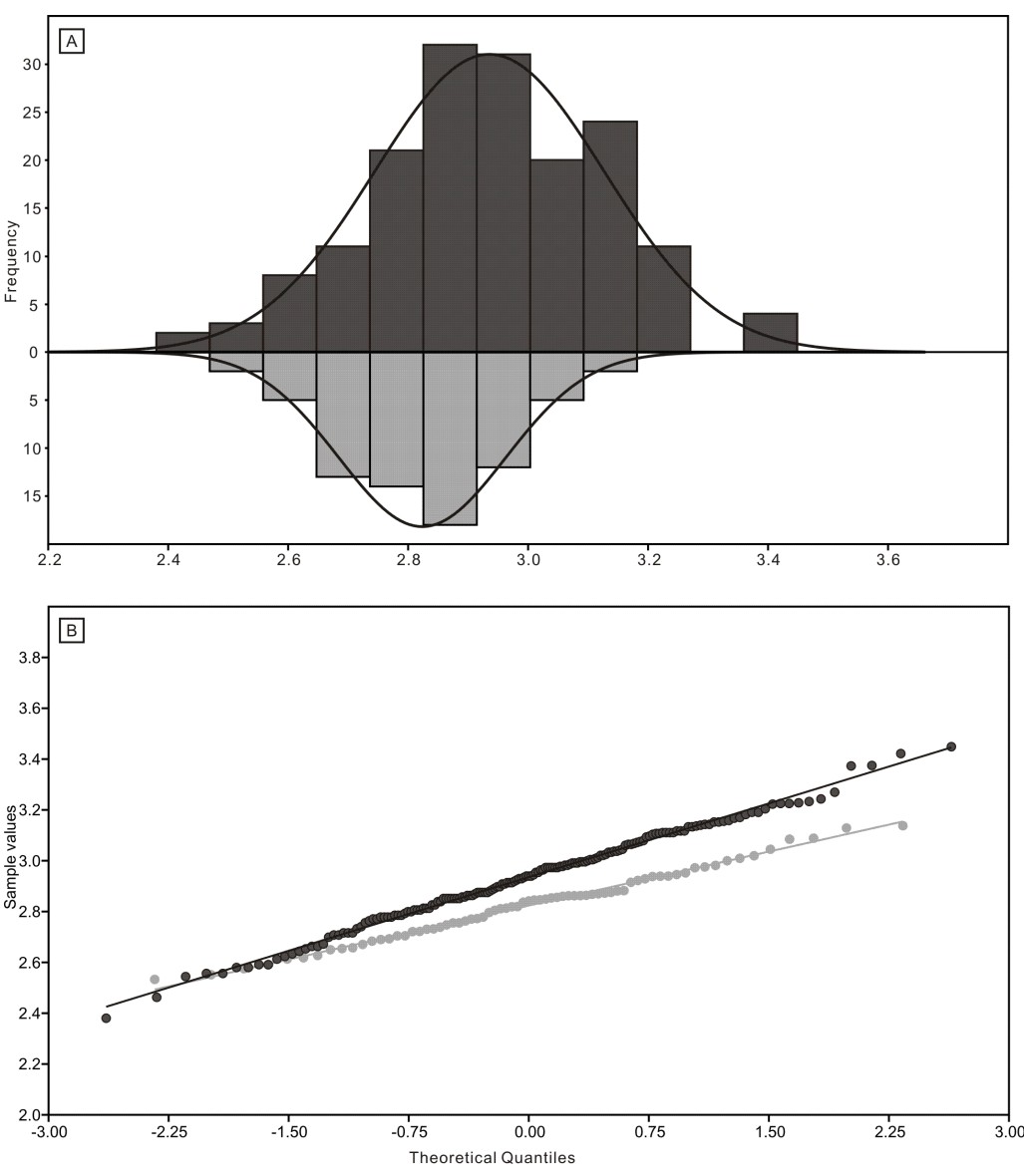

**Figure 7** **Distributions of length (after logarithmisation) for all conodont elements from the bedding planes and residues after acid-dissolving.** (A) Distributions of length. (B) Quantile-Quantile plot of the length.

significant decrease, while the ratio of M elements shows a slight or negligible decrease. This suggests that S elements have been more affected by the acid-dissolving method. Through isolated conodont elements obtained *via* the the acid-dissolving method, *Koike (2016)* proposed the apparatus compositions of five species of Ellisonidae, and his materials also showed that their M and S elements were more readily (but not better) preserved than P elements (Fig. 3), although his results could have been obscured due to the differences in size and shape of conodont elements (*Broadhead, Driese & Harvey, 1990*).

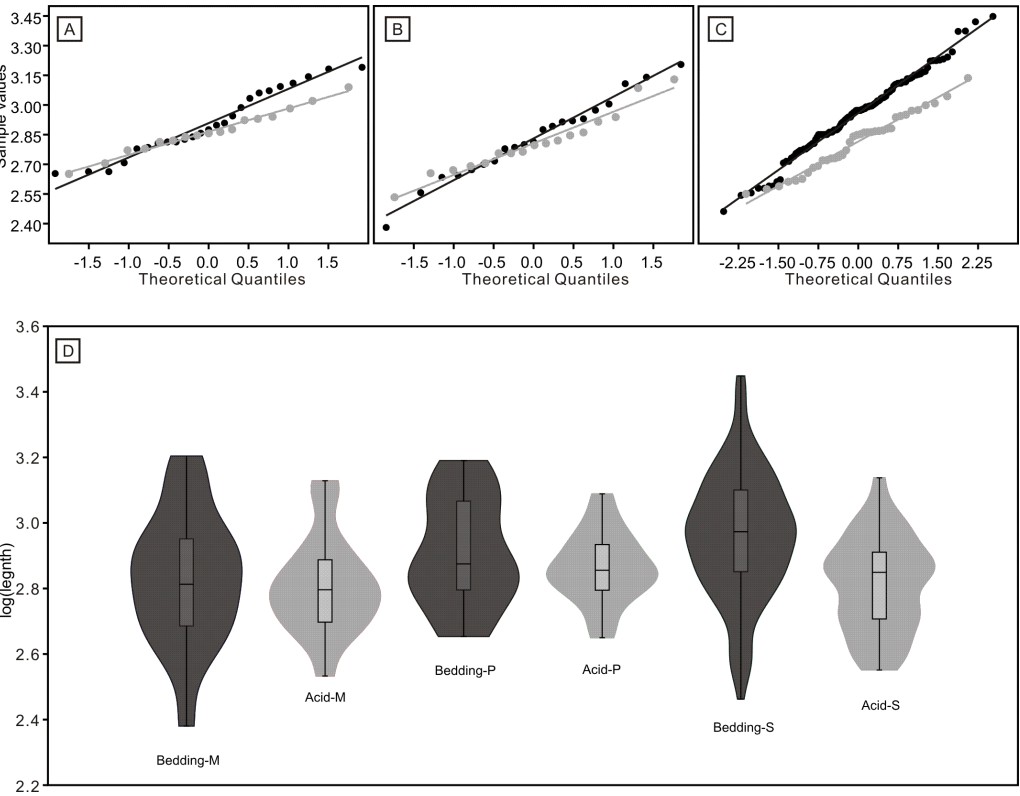

**Figure 8** **Distributions of length (after logarithmisation) for different conodont elements from the bedding planes and residues after acid-dissolving.** The dark dots represent data from the bedding planes. The grey dots represent data from the acid-dissolving method. (A) Distributions of length of P elements. (B) Distributions of length of M elements. (C) Distributions of length of S elements. (D) Violin-plot of different conodont elements.

The length distributions of conodont element from the two methods suggest that their preservation is affected by multiple factors (Table 1 and Figs. 8 and 9). Before being affected by the acid-dissolving process, M elements from the bedding-planes are smaller on average than P and S elements, while S elements are the largest among them. This is different from some reported well-preserved assemblages of Ellisonidae, which showed that P elements are smaller than M elements and that S elements are the largest (*Sun et al., 2020*), suggesting that M elements of Ellisonidae are more fragile than P elements. Additionally, research on the genus *Idiognathodus* showed that their S and M elements were usually larger than their P elements (see Fig. 4 in *Purnell, 1993*), which is consistent with Ellisonidae. As stated by *Orchard (2005)*, conodont elements exhibited a higher representation of pectiniform elements (usually P elements) when they were acquired from relatively nearshore, high-energy deposits where bias arising from post-mortem sorting and selective destruction cannot be ignored. This might be explained by element heterogeneity in mineralization or by their morphologies, as M elements are breviform digyrate and bear two inclined downward processes, while P elements are crescent-shaped angulate (*Sun et al., 2020*). S elements are smaller on average than P and M elements in the materials

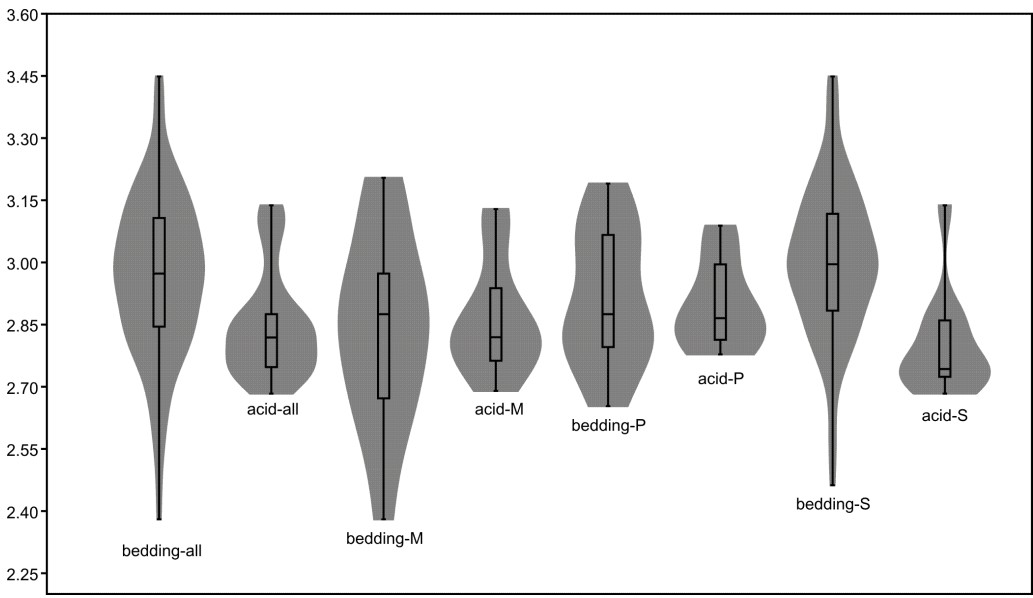

**Figure 9  Violin-plot of length of completely preserved conodont elements.**

acquired from the acid-dissolving method, and the other types also show reductions in size by eliminating larger individuals (Figs. 8 and 9). This suggests that conodont elements are influenced by the method, potentially leading to breakage, even for the less vulnerable P elements. Notably, elements in the same position of different conodont species have variable endurances. For example, a Middle Triassic multi-element research of *Nicoraella germanica* indicated that P and M elements are over-represented (Table 1 in *Chen et al., 2019*).

Previous studies have shown that the size of the conodont element is an ideal proxy for ecological changes (*Balter et al., 2008*; *Luo et al., 2008*; *Chen et al., 2013*; *Leu, Bucher & Goudemand, 2019*; *Wu et al., 2019*; *Zhang et al., 2020*; *Girard et al., 2023*). For example, diametrical or harmonious size-changing curves of conodont elements have been connected to transient or long-term ecological changes (*Chen et al., 2013*; *Ginot & Goudemand, 2019*; *Zhang et al., 2020*). However, conodont elements may exhibit different size variation trends during the same interval (*Leu, Bucher & Goudemand, 2019*). This might result from their different response mechanisms, which are further connected to their different habitats (*Joachimski et al., 2012*; *Sun et al., 2012*; *Leu, Bucher & Goudemand, 2019*; *Chen et al., 2021*). Although the size of conodont element can be controlled by ecological factors, and bias from laboratory processes has a limited impact on conclusions during conodont apparatus reconstructions (*Chen et al., 2016b*), it is still worth noticing that different degrees of influences may occur when data are used for different aims (*Jeppsson, 2005*). This study showed that conodont elements might have experienced different degrees of artificial damage during laboratory processes. Therefore, attention must be paid when trying to decipher conodont data for taxonomy, ecology, and other purposes, especially when conodont species have variant morphology of multi-elements.

## CONCLUSIONS

Conodont elements (including clusters) (Ellisonidae) from the bedding planes of the Early Triassic Nanzhang-Yuan'an Lagerstätte, as well as conodont elements acquired from the corresponding bed through the acid-dissolving method, provide insight into the biases that must be taken in account when deciphering conodont materials. Conodont elements from both methods exhibit varying degrees of bias, especially those from the acid-dissolving method, which introduces additional bias beyond that inherent to the bedding-plane materials. Owing to their different tolerances caused by different morphologies, conodont elements of Ellisonidae in different positions exhibit selective preservation or varing degrees of destruction even before laboratory processes. The widely used acid-dissolving method increases the bias by selectively destroying the M and S elements. Large individuals of all three different elements are prone to breaking during laboratory processing, with S elements being the most affected. This study indicates that biases in the size and morphology of conodonts caused by natural and artificial laboratory processes must be considered when deciphering these data.

## ACKNOWLEDGEMENTS

SEM pictures were taken at the State Key Laboratory of Biogeology and Environmental Geology in China University of Geosciences.

### Funding

This study is supported by the National Natural Science Foundation of China (grant Nos. 42102011, 42030513, 41972014), the Hubei Provincial Natural Science Foundation (2024AFD394), and the Science and Technology Special Fund of Hubei Geological Bureau (grants KJ2022-1, KJ2022-5). The funders had no role in study design, data collection and analysis, decision to publish, or preparation of the manuscript.

### Grant Disclosures

The following grant information was disclosed by the authors:
National Natural Science Foundation of China: 42102011, 42030513, 41972014.
Hubei Provincial Natural Science Foundation: 2024AFD394.
Science and Technology Special Fund of Hubei Geological Bureau: KJ2022-1, KJ2022-5.

### Competing Interests

The authors declare there are no competing interests.

### Author Contributions

- Kui Wu conceived and designed the experiments, performed the experiments, analyzed the data, prepared figures and/or tables, authored or reviewed drafts of the article, and approved the final draft.

- Boyong Yang conceived and designed the experiments, analyzed the data, prepared figures and/or tables, authored or reviewed drafts of the article, and approved the final draft.
- Bi Zhao conceived and designed the experiments, performed the experiments, prepared figures and/or tables, and approved the final draft.
- Liangzhe Yang conceived and designed the experiments, prepared figures and/or tables, and approved the final draft.
- Yarui Zou performed the experiments, prepared figures and/or tables, and approved the final draft.
- Gang Chen performed the experiments, prepared figures and/or tables, and approved the final draft.
- Jiangli Li performed the experiments, prepared figures and/or tables, and approved the final draft.

## Data Availability

The data is available in the Supplemental File.

## Supplemental Information

Supplemental information for this article can be found online at http://dx.doi.org/10.7717/peerj.18011#supplemental-information.

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
