# Peer review of "Discriminating conodont recording bias: a case study from the Nanzhang-Yuan’an Lagerstätte"

_PeerJ, doi:10.7717/peerj.18011_

## Round 0.1 · original submission · Major Revisions

You provide an interesting case study of potential biases in the record of Ellisonidae investigated with 2 different methods. The reviews are very disparate in their recommendations. The new data you provide is interesting, but the analyses as well as potential pitfalls needs to be better substantiated. The following crucial points need to be addressed before publication:

1) Research questions: define more clearly your research questions/hypotheses in the introduction (compare reviewer 2 and 3). What do you expect (e.g., is your hypothesis) concerning relative proportions of elements as well as size distributions.

2) Analyses: your data is interesting and novel, but could be more abundant (compare reviewer 3). Therefore, it is even more crucial to offer more appropriate analyses and visualization fo your results (compare reviewer 1). Your approaches also need to be more clearly explained (compare reviewer 2). Comparative analyses with other datasets (of the same group) would be useful keeping in mind potential pitfalls. Beyond reviewer suggestions (compare reviewer 1), it might be worth to consider plotting size distributions with NMDS and/or analyze size differences using linear mixed modelling particularly when you include additional datasets from the literature (e.g., Bhattacherjee et al. 2021, De Baets et al. 2022).

3) Data available: the raw measurement data needs to be made available for the sake of scientific reproducibility (compare reviewer 3)

4) Taxonomic identifications: I am not a Triassic conodont expert but I assume that even broken specimens of particular elements can sometimes be more precisely identified to species or at leas genus level (compare reviewer 1), so it is a bit peculiar to me that not more precise identifications were at least attempted. I can understand the merger of data on genus/species/family level due to limited availability but this strategy should be explicitly mentioned and its potential pitfalls discussed. As pointed out by reviewer 3, the relative abundances and associated size could also reflect true differences among taxa or the degree of breakage as opposed to preservational/preparation biases. Also differences in ontogenetic representation and random effects in your samples could play a role.

5) Interpretations: you attribute the patterns (different relative availability of particular elements and size distribution) you observe to the used preparation/study methods. In addition to more robust statistical analyses and ruling out alternative controls, interpreting the differences in your samples due to preparation/study methods would be more convincing if you address the underlying mechanisms more directly in your samples (e.g., does the size decrease relate to corrosion and/or breakage and/or sieving in the treated samples: compare reviewers 1 and 2). As pointed out by reviewer 3: What is the expected ratio of P, M, and S elements? and hence what technique is the most accurate? It might be more clear if the same samples were used – once for surface study and once for dissolution.

6) Pitfalls in interpretation: I feel your data is interesting and appropriate for study but could be more abundant, so potential pitfalls as well background information to interpret other factors controlling the patterns (e.g., depositional environments, homogeneity of conodont distribution in sediment matrix, alteration index, rock volume etc.) need to be more explicitly considered and discussed (compare suggestions by reviewers 1, 2 and 3). Both methods might be prone to biases which differ in direction or magnitude (compare reviewer 3).

7) References: I agree with reviewers 1, 2 and 3 that crucial references supporting particular statements need to be added. For example, the idea of biases in the conodont record related with depositional environments as well as methods of preparation is not new but crucial references are barely (e.g., Purnell and Donoghue 2005) or not discussed at all (compare reviewers 2 and 3). Additional references concerning size studies in conodonts (e.g., Girard et al. 2023) or more generally potential controls of smaller sizes in fossil samples (e.g., Mancini 1978, Twitchett 2001)

8) Formatting and language issues: there are some typographical and language issues (see reviewers 2 and 3). In this context, I recommend to let someone fluent in English proofread the manuscript before resubmission.

9) Figures: I agree with reviewer 1 that Figure 3A is currently difficult to understand. In addition, Fig. 3A shows largely similar patterns. It would be more appropriate to supplement the figures with lines showing the expected proportions of elements (4-2-9) and size as well as plotting size distribution from your data with other from the same family / lineage from the literature which could be added to Figure 3B and/or in an NMDS visualization or cluster analysis.

Please address these as well as all other points raised including those in annotated pdfs.

Suggested references:

Bhattacherjee, M., Chattopadhyay, D., Som, B., Sankar, A. S., & Mazumder, S. (2021). Molluscan live-dead fidelity of a storm-dominated shallow-marine setting and its implications. Palaios, 36(2), 77-93.

De Baets, K., Jarochowska, E., Buchwald, S. Z., Klug, C., & Korn, D. (2022). Lithology controls ammonoid size distributions. Palaios, 37(12), 744-754.

Girard, C., Charruault, A. L., Dufour, A. B., & Renaud, S. (2023). Conodont size in time and space: Beyond the temperature-size rule. Marine Micropaleontology, 184, 102291.

Mancini, E. A. (1978). Origin of micromorph faunas in the geologic record. Journal of Paleontology, 311-322.

Purnell, M. A., & Donoghue, P. C. (2005). Between death and data: biases in interpretation of the fossil record of conodonts. Special Papers in Palaeontology, 73, 7-25.

Twitchett, R. J. (2001). Incompleteness of the Permian–Triassic fossil record: a consequence of productivity decline?. Geological Journal, 36(3‐4), 341-353.

**Language Note:** The review process has identified that the English language must be improved. PeerJ can provide language editing services - please contact us at [email protected] for pricing (be sure to provide your manuscript number and title). Alternatively, you should make your own arrangements to improve the language quality and provide details in your response letter. – PeerJ Staff

Reviewer 1 ·

Basic reporting

The English should be improved in some cases for a better understanding. These are marked in the attached pdf.

Reference list is almost complete, I suggested some further papers to cite. (see in pdf)

The structure of the article is as it should be. Figures are high quality. For Fig. 3A some modifications are suggested to make it easier to understand.

Raw data are available.

Results are presented in detail, but discussions should be made more concrete in some cases, because it is difficult to get the message you want to give. For example it was not quite clear for me what you mean by size reduction in acid-treated samples. Breakage? Corrosion? Please see my detailed comments and related questions in the pdf.

Experimental design

The research is original, the question is well defined, but somehow the answer is hard to catch. A more detailed discussion could solve this problem.

I made some comments on the statistical analysis suggesting what you should analyse in a different way (see pdf). Please consider these comments in your revision.

Methods are described sufficiently.

Validity of the findings

The subject of the study is an interesting experiment. I am not sure though that you drew all possible conclusions from the results of your findings. The task is hard, because there are not a lot of studies to compare with, but I think caution has to be taken when comparing your results with studies that were made on conodonts which were deposited through different processes in different environments. Conodont elements behave as heavy minerals and therefore larger elements will settle from a current or mass transport sooner than smaller or thinner elements. I understand that your environment was low-energy, but how can you be sure of the even distribution of different elements in the investigated bed of limestone?

In the conclusion the breakage of large elements are stated, but this is not clearly discussed in the main text. Is this what causes the bias? Even broken pieces can be determined, very often on the species level. So why would this cause a bias?

Annotated reviews are not available for download in order to protect the identity of reviewers who chose to remain anonymous.

·

Basic reporting

The english need to be improved, as well as some figures. I have indicated in the attached pdf, some references that can be added,

Experimental design

The idea of the existence of conodont bias is not new, but the authors try to statistically test it, so I think is a good approach. However I think that they need the explain better what they were comparing, in terms of rock volume, to be sure that both "methods " are comparable.

Validity of the findings

Although a relatively simple approach I think is appropiated, although I think that the authors need to explain better their results trying to analyis how the process is affecting the final results, are mechnical (breakage), disolition of some elements, they were losing elements because the sise of the shieve??

Additional comments

The manuscript deal with the always interesting, and hard, attempts to show the important bias that affect the condont fossil record and their posterior consequence for different palaeobiological studies. I think that although the topic is not new, the authors show numericaly the differences in rates and size of single conodont collection obtained by different methods. Although the idea and goals are clear, I think that the manuscript needs to be revised by a fluent English speaker. I´m not a native speaker and I cannot help, but it needs to be revised and improved.
I’ve made some comments in the main text, many of them suggestions or just clarification for the reader that the authors could address before the resubmission. But I think that they need to explain better how the different process are affecting the conodonts, not just indicating that there are differences, why this differences?
Because they need to improve the English and better identify the bias for a better explanation of their conclusion I recommend its publication with very moderate-major revision.

Reviewer 3 ·

Basic reporting

The English could be improved throughout, especially lines 30, 32, 36, 89-91, 106, 107, 129, 148-151, etc.
Citations are not always appropriate: the authors write a long sentence with several arguments, some very general and consentual, sometimes mixed with less consentual ones and they cite several 'classical' papers that do not directly relate to the various points of their sentence. See for instance, lines 31, 48 (see instead Purnell and Donoghue 2005, Between Death and Data: Biases in interpretation of the fossil record of conodonts, and the other contributions of the associated special volume 73 in Special Papers in Palaeontology), 161-164.

Raw data not shared.

Superficial results with no clearly stated hypotheses (what is the expected ratio of P, M, and S elements? and hence what technique is the most accurate?)

Experimental design

Research question not well defined. The research does not fill an identified knowledge gap. In fact we knew already that such biases exist and they have been already quantified in the past. The number of elements used here does not allow for strong conclusions anyway and several aspects have not been controlled : what species of Ellisonidae? (comparisons across taxa are expected to reflect the many biases already documented for relative abundances and contrastingly, differences in relative sizes do not necessarily reflect preservational biases but may reflect true differences among taxa), are the elements measured broken elements?

Validity of the findings

The conclusions are superficial and they are not supported by the data. Bedding plane assemblages show proportionally more S elements, but they also show proportionally less P elements. Which technique is the most accurate? that is, the closest to the expected ratios of elements?
The authors seem to neglect the fact that both techniques are prone to sampling biases but those biases are different from one another (mesh sizes of the sieved in the acid extraction method, minimal size for detection in the other).

---

## Round 0.2 · Minor Revisions

Thank you for addressing our suggestions. I agree with the reviewer that it has significantly broadened the scope, readability and reproducibility of your work. I look forward to seeing this work published; however, i feel some minor but crucial aspects still need to be addressed before publication. The main points to be addressed are:

New analyses and graphs: I greatly welcome the new analyses and I feel the new graphs more appropriately capture your results and make it easier to follow your interpretations. To understand the significance and potential impact of sampling, it is necessary to add sample size numbers of categories/samples explicitly in the graphs (e.g., Figs 2-4) as well as adding binomial error bars or confidence intervals in figures (Figs 3-4). For examples of binomial confidence intervals: see, for example, Raup (1991), De Baets et al. (2012; Fig. 5) or Takeda & Tanabe (2014; Fig. 9). For comparisons, relative proportions or percentages are more appropriate than absolute numbers (e.g., Fig. 2). Please clarify what the y-axis in Fig. 2 represents as well as “ref” category in Fig. 3.

Formatting and language issues: I discovered various issues with phrasing. Some relate to the use of terms in the wrong context, well other sentences could be misunderstood by the reader (see annotated pdf).

These and other points can be found in the annotated pdf.

I look forward to receiving your revised manuscript.

Suggested references:

De Baets, K., Klug, C., Korn, D., & Landman, N. H. (2012). Early evolutionary trends in ammonoid embryonic development. Evolution, 66(6), 1788-1806.

Raup, D. M. (1991). The future of analytical paleobiology. Short Courses in Paleontology, 4, 207-216.

Takeda, Y., & Tanabe, K. (2014). Low durophagous predation on Toarcian (Early Jurassic) ammonoids in the northwestern Panthalassa shelf basin. Acta Palaeontologica Polonica, 60(4), 781-794.

Reviewer 1 ·

Basic reporting

The authors made a very thorough review and tried to consider all reviewers' comments and suggestions. The previously ambiguous parts of the text are now made clear and the modified figures reflect the results in a better way.
The scientific problem of the paper is quite interesting and the results might have a great impact on future conodont studies.

Experimental design

no comment

Validity of the findings

no comment

---

## Round 0.3 · Minor Revisions

Thank you for adding the confidence intervals which make your study even easier to follow and reproduce. Your manuscript is as good as accepted. There are just 2 minor issues which need to be resolved before publication:

1) Completeness of the reference list: One reference cited twice in text is missing from the text (see annotated pdf). Please verify once more that all references in the text are also cited in the reference list.
2) Confidence intervals: the addition of the binomial confidence intervals makes it easier to evaluate your results and reproduce your interpretations. However, there are no references cited for its use. Please make sure reference(s) are cited which approach you are following (e.g., some possibilities were mentioned in the previous decision; compare annotated pdf). In addition, please (briefly explain in the caption to Figure 3 why no confidence intervals are plotted for 3K.

I am looking forward to seeing this research published.

---

## Round 0.4 · accepted · Accept

Thank you for addressing these final suggestions which make the manuscript even easier to follow and of broader relevance. There remain some minor typographical errors ("Baets, K. D." should be "De Baets, K.": see annotated pdf). Please make sure to address these typographical issues during the proofing phase.